# Genomic Scan for Runs of Homozygosity and Selective Signature Analysis to Identify Candidate Genes in Large White Pigs

**DOI:** 10.3390/ijms241612914

**Published:** 2023-08-18

**Authors:** Chang Yin, Yuwei Wang, Peng Zhou, Haoran Shi, Xinyu Ma, Zongjun Yin, Yang Liu

**Affiliations:** 1Department of Animal Genetics and Breeding, College of Animal Science and Technology, Nanjing Agricultural University, Nanjing 210095, China; 2021105019@stu.njau.edu.cn (C.Y.); 2022805132@stu.njau.edu.cn (Y.W.); 2022105032@stu.njau.edu.cn (P.Z.); 15119306@stu.njau.edu.cn (H.S.); maxinyu@stu.njau.edu.cn (X.M.); 2College of Animal Science and Technology, Anhui Agricultural University, Hefei 230036, China; yinzongjun@ahau.edu.cn

**Keywords:** pig, genomic, selective signature, ROH

## Abstract

Large White pigs are extensively utilized in China for their remarkable characteristics of rapid growth and the high proportion of lean meat. The economic traits of pigs, comprising reproductive and meat quality traits, play a vital role in swine production. In this study, 2295 individuals, representing three different genetic backgrounds Large White pig populations were used: 500 from the Canadian line, 295 from the Danish line, and 1500 from the American line. The GeneSeek 50K GGP porcine HD array was employed to genotype the three pig populations. Firstly, genomic selective signature regions were identified using the pairwise fixation index (F_ST_) and locus-specific branch length (LSBL). By applying a top 1% threshold for both parameters, a total of 888 candidate selective windows were identified, harbouring 1571 genes. Secondly, the investigation of regions of homozygosity (ROH) was performed utilizing the PLINK software. In total, 25 genomic regions exhibiting a high frequency of ROHs were detected, leading to the identification of 1216 genes. Finally, the identified potential functional genes from candidate genomic regions were annotated, and several important candidate genes associated with reproductive traits (*ADCYAP1*, *U2*, *U6*, *CETN1*, *Thoc1*, *Usp14*, *GREB1L*, *FGF12*) and meat quality traits (*MiR-133*, *PLEKHO1*, *LPIN2*, *SHANK2*, *FLVCR1*, *MYL4*, *SFRP1*, *miR-486*, *MYH3*, *STYX*) were identified. The findings of this study provide valuable insights into the genetic basis of economic traits in Large White pigs and may have potential use in future pig breeding programs.

## 1. Introduction

Large White pig breeds have gained recognition for their exceptional performance, characterized by rapid growth, efficient feed conversion, and high-carcass yield [1]. Through a combination of natural and artificial selection, these pigs have not undergone a lot of different and significant evolutionary changes, with the recent emphasis on strong and directional positive selection, aligning their characteristics more closely with human requirements. The advent of cost-effective, high-throughput sequencing techniques has facilitated comprehensive genome-wide analyses of genetic structure and relationships within animal populations. Notably, the application of the fixation index (F_ST_) and locus-specific branch length (LSBL) methodologies have emerged as a valuable approach for discerning selection signatures and distinguishing traits across diverse breeds and geographic regions [2,3,4].

In addition to SNP and gene expression data, runs of homozygosity (ROH) have emerged as valuable components of omics data available in biological databases, serving as powerful tools for gene discovery and diversity assessment in livestock. ROH refers to contiguous segments of the genome where an individual inherits identical haplotypes from both parents [5,6]. Long haplotype fragments are derived from a closer common ancestor, whereas short haplotype fragments are derived from a distantly related common ancestor [7]. Several factors can influence the development of ROH patterns on the genome, such as inbreeding, genetic drift, the mating system, selection intensity, effective population size, population structure, and genetic linkage [8]. In 1999, early studies revealed that the length of homozygous fragments is associated with human diseases, underscoring their importance [9]. Today, the advent of high-throughput sequencing techniques enables convenient access to genomic information. The use of high-density SNP markers for scanning the genome were proposed to identify regions with reduced heterozygosity, thus enabling ROH detection [10]. The widespread adoption of SNP chips and whole-genome resequencing offers excellent opportunities to investigate ROH in livestock. Recent studies in pigs have employed ROH to explore signatures of selection. Wu et al. [11] found some candidate selection signatures within the DSE pig population were detected through the ROH islands. Wang et al. [12] used Duroc (American origin) and Duroc (Canadian origin) pigs to investigate the harmful ROH regions on five economic traits. Xu et al. [13] described the occurrence and distribution of ROH in the genome of Jinhua pigs and found several genes within ROH.

In this study, we utilized the GeneSeek 50K GGP porcine HD array to characterize Canadian, American, and Danish Large White pigs. F_ST_ and LSBL were employed as methods to select signatures associated with specific traits. Additionally, homozygosity analysis was conducted to further investigate the genetic characteristics of the three pig populations. The findings of this research contribute novel and valuable insights into the population history and genetic structure of Large White pigs with diverse genetic backgrounds.

## 2. Results

### 2.1. Population Stratification Assessment

Principal component analysis was conducted to assess the genetic variation indices of the three pig populations. The results demonstrated significant differences in the genetic backgrounds of the American, Canadian, and Danish line Large White pigs, with PC1 effectively separating the three populations (Figure 1).

### 2.2. Selective Signature Analysis

To minimize the risk of false positive selection signals, it has become common practice to employ multiple detection methods, allowing for cross-validation and strengthening the reliability of the findings. In this study, we utilized two such methods, namely F_ST_ and LSBL, for selection signature detection. Through a comprehensive genome-wide selective sweep analysis, we identified a total of 1434 enriched genes after filtering the intersection of the top 1% windows obtained from F_ST_ (Figure 2A) and LSBL (Figure 2B). Furthermore, we observed that 83 loci overlapped between these two methods (Figure 2C), indicating a mutual validation of the results. Candidate genes located within genomic regions with high frequencies of F_ST_ and LSBL are shown in Table 1.

In each selection signature analysis, the top 20% of genomic regions were identified and subjected to further investigation. Functional annotation of all genes residing within these selected genomic regions was performed using the WebGestaltR package [14]. Subsequently, significant enrichment was observed in specific functional categories within each method. In the LSBL analysis (Figure 3B), the genes exhibited significant enrichment in processes related to morphogenesis of an epithelium, plasma membrane protein complex, transcription factor complex, oxidoreductase activity involving the CH-OH group of donors with NAD or NADP as acceptors, and cytoskeletal protein binding. Conversely, in the F_ST_ analysis (Figure 3A), the genes displayed significant enrichment in peptide metabolic processes, vesicle-mediated transport, catalytic complex, and protein-containing complex binding.

Finally, the results of the KEGG pathway analysis showed that MAPK signalling pathways were significantly enriched in FST and LSBL (Figure 3C,D). The MAPK signal pathway is involved in skeletal muscle regeneration [15]).

### 2.3. Runs of Homozygosity Analysis

The genome-wide ROHs were assessed on 18 autosomes of all tested individuals. After the read filtering procedures, 34,150, 34,543 and 34,497 SNPs and 500, 295 and 1500 individuals were retained from the Canadian line, Danish line and American line, respectively. These SNPs were retained for subsequent ROH analysis. 

The association between the total genomic length covered by runs of homozygosity (ROH) per individual and the total number of ROH per individual was examined and presented in Figure 4A. The Danish line displayed a higher number of ROH compared to the Canadian and American lines. Furthermore, within the Danish line, certain individuals exhibited exceptionally long ROH segments covering more than 750 Mb. Analysis of autosomes (Figure 4B) revealed variations in the number of ROHs across the three populations, indicating an uneven distribution of ROHs. Interestingly, all three groups exhibited the fewest ROHs per chromosome on SSC 18, while the highest number of ROHs was observed on SSC 1. The distribution of ROH according to length is shown in Figure 4C. To further assess the distribution of ROHs on autosomes, we estimated the coverage of ROHs for each autosome. Notably, SSC10 displayed the lowest coverage of ROH segments (Figure 4D).

To identify the genomic regions most associated with ROH in the three pig populations, we calculated the frequency of SNP occurrence within the ROH segments. From these calculations, we selected the top 1% of SNPs with the highest frequency and plotted their positions along the respective chromosomes (Figure 5A–C). Our analysis revealed a total of 25 genomic regions exhibiting a high frequency of ROH, encompassing a range of lengths from 7.9 kb on SSC6 to 5.12 Mb on SSC6, as presented in Table 2. On chromosome 6, from position 105,105,811 to 107,369,304, there is only a shared overlap in the ROH of these three populations. Furthermore, 12% of the total ROH length was discovered in the American lines; 11% in the Canadian lines; and 14% in the Danish lines. The longest ROH segment was identified on SSC6, spanning 121 SNPs. In Table 2, a comparison between the identified QTLs in this study and those catalogued in pigQTLdb reveals noteworthy associations. Specifically, in the Danish line, a genomic segment spanning 54.09 Mb to 54.25 Mb on SSC5 is linked to reproductive traits. In the American line, the chromosomal region from 14.60 Mb to 14.90 Mb onSSC1 is associated with average daily weight, while the region between 72.34 Mb and 74.40 Mb on SSC7 is correlated with fat area percentage in the carcass. In the Canadian line, the genomic region ranging from 44.73 Mb to 448.38 Mb on SSC4 is implicated in ham weight, and the span from 70.70 Mb to 74.26 Mb on SSC7 is tied to the teat numbers. These findings indicate that the selection emphasis within the Canadian line is centred on reproductive traits and growth performance. Moreover, we identified 1216 genes within these ROH regions. Intriguingly, our analysis uncovered 32 overlapping genes across all three populations of ROH (Figure 5D). These findings further enhance our understanding of the genetic architecture and potential selective pressures acting upon these regions. Candidate genes located in genomic regions with high frequencies of ROH are shown in Table 3.

The 1216 candidate genes identified within ROH in pigs were subjected to functional enrichment analyses. Gene Ontology (GO) analysis revealed significant enrichment of specific biological processes (Figure 6A). Kyoto Encyclopedia of Genes and Genomes (KEGG) enrichment analysis was performed to further elucidate the pathways associated with these genes, as depicted in Figure 6B–D. In the American line, the candidate genes were significantly enriched in processes related to muscle cell differentiation, muscle system function, U12-type spliceosomal complex, as well as oxidoreductase activity involving the CH-OH group of donors with NAD or NADP as acceptor. In the Canadian line, significant enrichment was observed in processes related to muscle hypertrophy, regulation of G protein-coupled receptor signalling pathway, muscle adaptation, zinc ion binding, and oxidoreductase activity involving the CH-CH group of donors with NAD or NADP as acceptor. In the Danish line, significant enrichment was observed in processes related to tumour necrosis factor receptor binding, lipid droplets, and negative regulation of gliogenesis.

The KEGG pathway analysis demonstrated the enrichment of specific pathways associated with the candidate genes identified in each pig population. In the Danish line, candidate genes were significantly enriched in the PPAR signalling pathway and the cGMP-PKG signalling pathway (Figure 6D). In the Canadian line, candidate genes showed significant enrichment in the Spliceosome pathway and the one-carbon pool by the folate pathway (Figure 6B). In the American line, candidate genes were notably enriched in the Regulation of the actin cytoskeleton pathway and the Focal adhesion pathway (Figure 6C).

## 3. Discussion

The identification of numerous candidate genes associated with economic traits in this study contributes to our understanding of the genetic factors influencing these traits in pigs (Table 1). Among the identified genes, *MYL4* was found to exhibit differential expression in the longest muscle tissue of the pig’s back, which correlated with variations in the number of muscle fibres within the tissue [16]. Other genes of interest include *FGFR2*, which has been recognized as a crucial regulator of myogenesis during skeletal muscle development and regeneration [17], and *SFRP1*, predicted to be targeted by miRNA-1/206, and implicated in muscle cell proliferation and prenatal skeletal muscle development [18]. *SHANK2*, a member of the Shank protein family, was found to be associated with childhood obesity, and to influence oestradiol blood concentration [19,20]. Induction of miR-486 takes place during the differentiation of myoblasts, whereby they directly target the 3′ untranslated region (UTR) of Pax7 leading to its downregulation. This downregulation mechanism promotes the differentiation of muscle cells [21]. The *MYH3* gene, encoding myosin heavy chain 3, was identified as a regulator of myofiber-type specification and adipogenesis in skeletal muscle [22]. In a study by Lin et al. [23], the *SHANK2* gene was identified as likely to affect the backfat thickness in pigs. *FLVCR1* deficiency results in Diamond–Blackfan anaemia, often associated with skeletal malformations [24]. *STYX*, the main signalling regulator in the ERK1/2 MAPK signalling pathway, increased pre-adipocyte adipogenesis by promoting pre-adipocyte proliferation [23]. Knockout *LRRK2* displayed lipid accumulation in the liver and kidney of rodents. *LRRK2* was identified as an important gene for Intramuscular fat content (IFC) using GWAS in Suhuai pigs [25]. *FGF12* exhibits significant and localized expression in midgestation mouse embryos and plays a crucial role in inducing the differentiation of mouse embryonic stem cells [26]. The protein encoded by *CD247* is T-cell receptor zeta, which has an important role in antigen recognition and signal transduction [27].

In this study, the GeneSeek 50K GGP porcine HD array was utilized to investigate the frequency and distribution of ROH in the genome of three distinct pig breeds. The formation of ROH patterns is primarily influenced by various factors, including population bottlenecks, inbreeding, genetic drift, and selective pressures arising from both natural and artificial selection [28]. Our findings revealed variations in the number and coverage of ROHs among chromosomes, with a general trend of increasing ROH numbers along with the length of the chromosome. Interestingly, shared ROHs among individuals in livestock populations may not solely be attributed to demographic history but could also reflect selection pressures [8]. Therefore, exploring ROH islands can provide valuable insights into potential selection signatures and shed light on different selection events, the direction of selection, and adaptations to diverse production systems [29]. Regarding our results, the ROH lengths observed in the three populations were approximately 500 Mb, and no significant differences were found among the three populations in terms of ROH characteristics. These findings contribute to our understanding of the genomic landscape of ROH and provide important implications for the genetic selection and adaptation of pig populations in various production contexts.

The present study identified distinct sets of candidate genes from both selection signature analysis and ROH analysis. In pigs, inbreeding represents a combination of natural and artificial events, and our results demonstrate the complementary nature of different methods in investigating complex traits. There is some difference in the focus of artificial selection in these three distinct Large White Pigs discussed in this study. The Danish line focuses on reproductive performance, such as litter size; the American line focuses on growth performance, such as growth rate and meat ratio; and the Canadian line is similar to the American line in that it also focuses on growth performance. In order to improve the populations, all three populations were subjected to accurate phenotypic data collection and were regularly monitored, and data were analysed to identify the best-performing breeding individuals. The identification of candidate genes associated with economic traits was based on genomic regions exhibiting a high frequency of ROH. Functional analysis and previous studies support the association of most candidate genes with economic traits (Table 3). Several candidate genes relating to reproduction traits were identified: *ADCYAP1* global knockout has decreased fertility and affects spermatogenesis [30,31]. *U2* and *U6* play an important role in snRNP assembly and pre-mRNA splicing in oocytes. An essential role for *CETN1* is in the late steps of spermiogenesis and spermatid maturation, and this gene plays a role in the reproductive capacity of the Danish line [32]. Loss of spermatocyte viability is a consequence of defects in the expression of genes regulation by *THOC1* required, which means that this gene also has an effect on the ability of the Danish line to reproduce [33]. *USP14* is required for spermatid differentiation during spermiogenesis [34]. The knockdown of *GATA6* resulted in a loss of the normal steroidogenic testis function [35]. *GREB1L* plays a major role in genital development [36]. In addition, Niu et al. [37] claim that *GREB1L* were potential candidate genes for controlling the expression of the rib number. Some genes associated with specific traits related to meat quality were detected: *MiR-133* repressed ERK1/2 activity by targeting *FGFR1* and *PP2AC* to repress myoblast proliferation and promote its differentiation [38]. *PLEKHO1* depletion drastically impairs C2C12 myoblast fusion in vitro and in vivo during zebrafish muscle development [39]. *LPIN2* is one member of the lipid gene family associated with backfat thickness in pigs [40]. This gene is located in the genomic region of the high frequency ROH gene in the American line in this study, suggesting that it is related to the growth performance related to backfat thickness.

## 4. Materials and Methods

### 4.1. Ethics Statement

The Animal Welfare Committee of Nanjing Agricultural University conducted a review of all animal testing and sample collection techniques used in this research. This review process included a careful examination of the ethical considerations of the research, as well as the methods and procedures used to ensure the safety and welfare of the animals involved. The Committee approved the animal testing and sample collection techniques in this research, ensuring that the animals were treated humanely, and that the data collected were accurate and reliable. (Permit number: DK652).

### 4.2. DNA Sampling and Sequencing of DNA

This study utilized three distinct populations of Large White pigs, including 500 Canadian (CLW, which were from Chongming county in Shanghai), 295 Danish (DLW, which were from Huaibei city in Anhui), and 1500 American (ALW, which were from Lixin county in Anhui) Large White pigs, as experimental materials. Genomic DNA was extracted from ear tissue and genotyped with the GeneSeek 50K GGP porcine HD array. The software PLINK (V1.90) (http://www.cog-genomics.org/; accessed on 16 March 2023) [41] was used for quality control of the data and the following standards were set: (i) removal of SNP loci with a call rate of less than 0.95 and unknown positions; (ii) removal of SNP loci with a minor allele frequency (MAF) of less than 0.05; and (iii) discarding of individuals with a call rate of less than 0.95. SNP genome coordinates were obtained from the *Sus scrofa* 11.1 porcine genome reference assembly.

### 4.3. Population Structure

Principal component analysis (PCA) was performed using PLINK1.9 [41], and the results of structure and PCA were visualized using the R package “barplot” and “ggplot2”, respectively [42].

### 4.4. Partitioning Heritabilities of Complex Traits Based on Selection Signatures

The F_ST_ method based on population differentiation was used to analyse the selection signature of the data of the three pig populations. The F_ST_ was calculated with VCFtools [43] (—fst-window-size 50,000—fst-window-step 10,000). LSBL (L_CLW_, L_DLW_, L_ALW_) were calculated from single locus pairwise F_ST_ distances, where L_CLW_ = (CLW-DLW F_ST_ + CLW-ALW F_ST_ − DLW-ALW F_ST_)/2, L_DLW_ = (CLW-DLW F_ST_ + DLW-ALW F_ST_ − CLW-ALW F_ST_)/2 and L_ALW_ = (CLW-ALW F_ST_ + DLW-ALW F_ST_ − CLW-DLW F_ST_)/2 [44]. Based on the annotation file of the reference genome, the top 1% of the selected loci were screened.

### 4.5. Runs of Homozygosity Detection

ROH were detected with the detect RUNS package of R software version 4.0.5; we defined ROH according to the following criteria: (i) the minimum number of SNPs in a sliding window was 50; (ii) one heterozygous genotype and no more than two missing SNPs were allowed per window; (iii) the minimum ROH length was set to 1 Mb to eliminate the impact of strong linkage disequilibrium (LD); (iv) the minimum SNP density was 1 SNP every 500 kb and the maximum gap between consecutive SNPs was set to 1 Mb to avoid affecting the length of ROH with a low SNP density; and (v) to minimize the number of the false-positive ROH, the minimum number of SNPs that constituted the ROH (l) was calculated with the method proposed by [45], $I=\frac{ln\alpha /\left(n_{s}\times \;n_{i}\right)}{ln\left(1-het\right)}$, where α is the percentage of false-positive ROH, $n_{s}$ is the number of SNPs per individual, $n_{i}$ is the number of individuals and het  is the proportion of heterozygosity across all SNPs.

In this study, the detected ROHs were divided into three categories for further analysis: 1–5, 5–10, and >10 Mb. We computed the frequency of ROH numbers and the average length of an ROH per breed.

### 4.6. Detection of Common Runs of Homozygosity

We calculated the frequency of occurrences within the ROH regions of each SNP across the individuals and made a Manhattan figure by plotting these values in conformity with the position of each SNP on chromosomes. The SNPs in the top 1% of the frequency of occurrence were selected as a hint of a potential ROH.

### 4.7. Pathway and Functional Analysis

Candidate genes were annotated via the Ensembl database (*Sus scrofa* 11.1, http://www.ensemble.org/; accessed on 16 March 2023) at 100-kb regions (upstream 50 kb and downstream 50 kb) flanking the SNPs of ROH hotspots. The Gene Ontology (GO) terms and Kyoto Encyclopedia of Genes and Genomes (KEGG) pathways were analysed for all candidate genes by the Metascape database (https://metascape.org/; accessed on 16 March 2023).

### 4.8. Gene Annotation

To determine positional candidate genes, we utilized the BioMart database (http://www.ensembl.org/) for annotating significant SNPs loci. In our study, candidate genes were identified within a 500-kb genomic region upstream and downstream of the significant SNPs. Furthermore, functional annotation of genes within the regions of interest was conducted using the R package WebGestaltR [14]. Additionally, an extensive literature review was performed to gather pertinent information on gene functions for exploratory investigations.

## 5. Conclusions

In this study, we investigated the selection signatures and runs of homozygosity (ROH) in three Large White pig populations (Canadian, Danish and American) from the porcine 50 K SNPs chip. Our analysis revealed several candidate genes associated with reproductive traits (*ADCYAP1*, *U2*, *U6*, *CETN1*, *Thoc1*, *Usp14*, *GREB1L*, *FGF12*) and meat quality traits (*MiR-133*, *PLEKHO1*, *LPIN2*, *SHANK2*, *FLVCR1*, *MYL4*, *SFRP1*, *miR-486*, *MYH3*, *STYX*) located within genomic regions exhibiting a high frequency of ROH and selection signatures. Our findings suggest that *GREB1L* may play a role in controlling the expression of rib numbers. These results provide valuable insights into the genetic basis of reproductive and meat quality traits in Large White pigs and contribute to our understanding of the molecular mechanisms underlying these economically important traits.

## Figures and Tables

**Figure 1 ijms-24-12914-f001:**
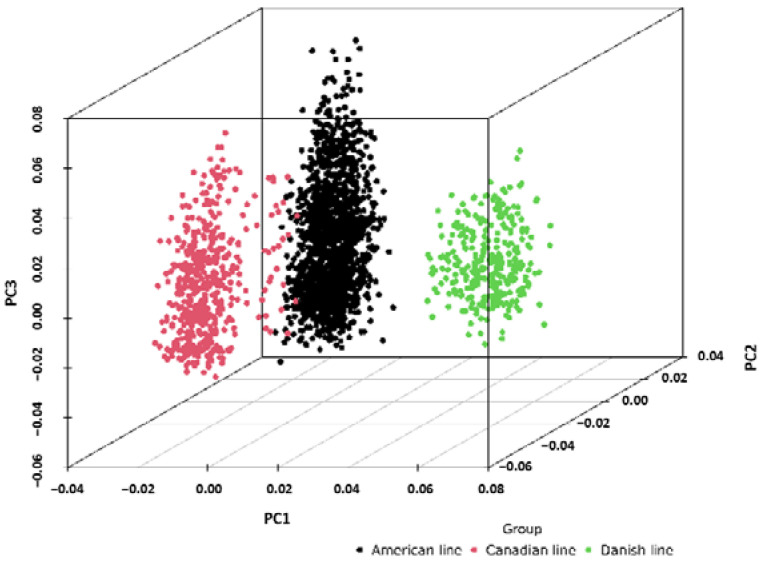
Principal component analysis (PCA) of three Large White pig populations.

**Figure 2 ijms-24-12914-f002:**
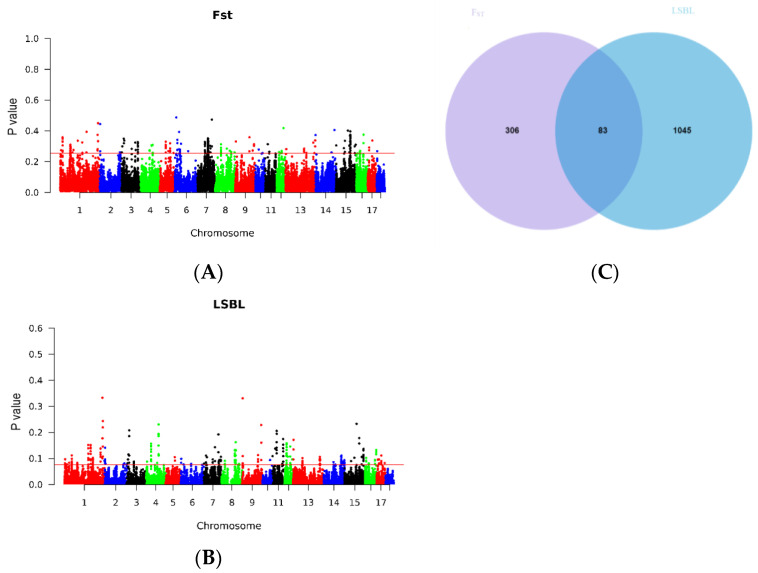
Genome-wide selective signature analysis. (**A**) Manhattan plots of the distribution of F_ST_ on the autosomal chromosomes calculated; (**B**) Manhattan plots of the distribution of LSBL on the autosomal chromosomes calculated; (**C**) Venn diagrams of the Top 1% of genes in F_ST_ and LSBL.

**Figure 3 ijms-24-12914-f003:**
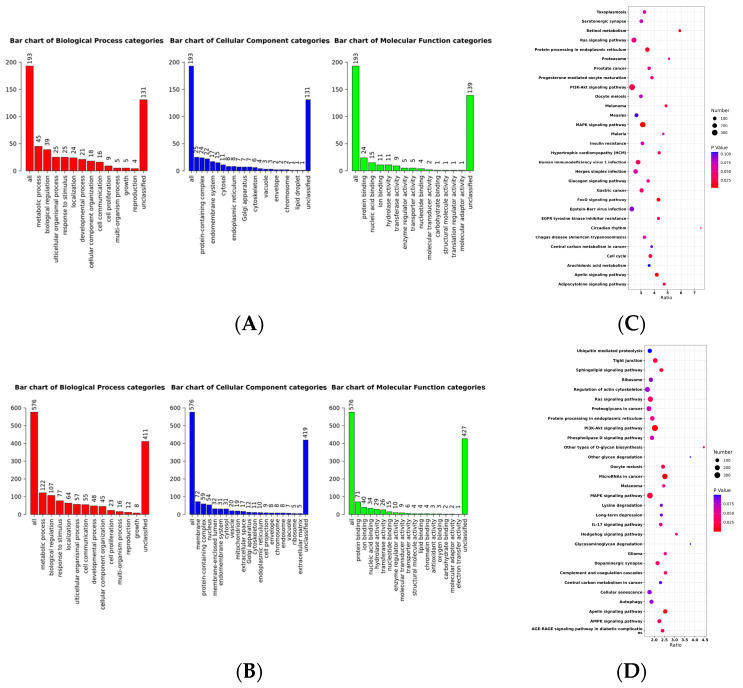
Kyoto Encyclopedia of Genes and Genomes enrichment analysis. (**A**) GO analysis of FST; (**B**) GO analysis of LSBL; (**C**) KEGG pathway analysis of FST; (**D**) KEGG pathway analysis of LSBL. In graphs (**A**,**B**), the abscissa represents the GO terms that were the most enriched; the ordinate represents the number of genes that were enriched in this classification. The size of the circles represents the number of genes contained in the particular class in the graph (**C**,**D**), the larger the circle is, the more genes there are. Differently coloured circles represent the enrichment degree of false positives, the redder the circle is, the lower the false positive rate.

**Figure 4 ijms-24-12914-f004:**
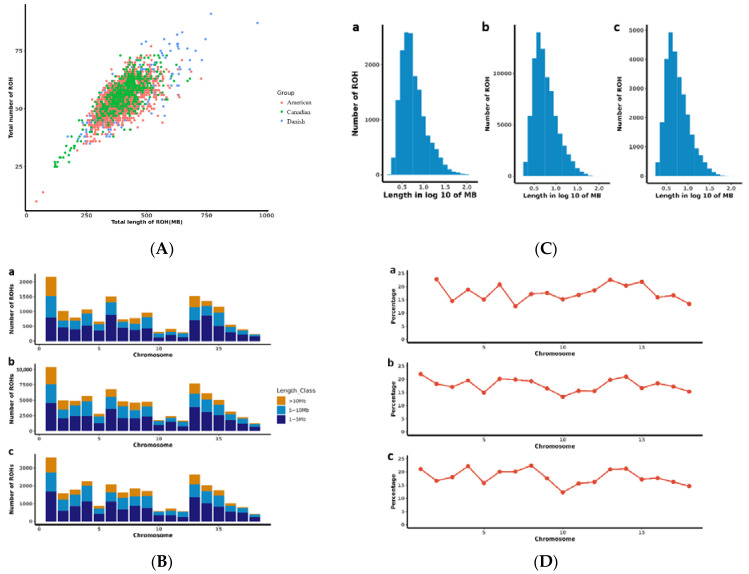
Distribution of runs of homozygosity. (**A**) Total genomic length (Mb) covered by ROH; (**B**) The number of ROH on each chromosome; (**C**) Distribution of ROH in different lengths (Mb). The values of length in Mb were transformed in log 10, a presents as the American line, b presents as the Canadian line, c presents as the Danish line; (**D**) The ROH coverage on each chromosome.

**Figure 5 ijms-24-12914-f005:**
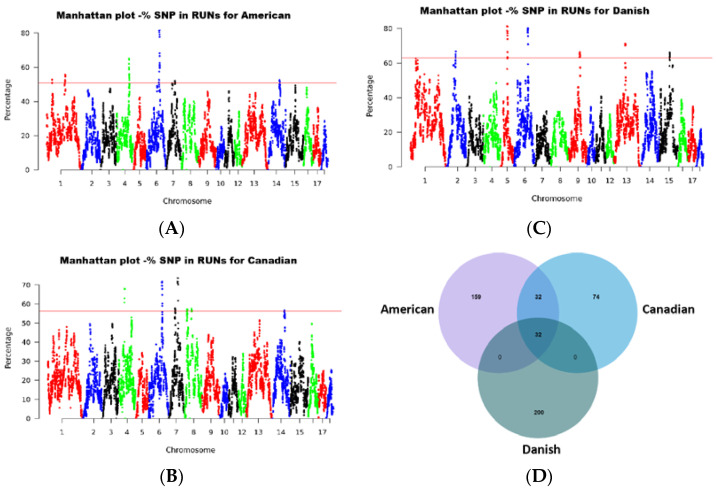
Frequency of occurrences of each SNP within ROH regions among all of the individuals. (**A**) American line; (**B**) Canadian line; (**C**) Danish line; (**D**) Venn diagrams of the top 1% of genes in three Large White pig populations.

**Figure 6 ijms-24-12914-f006:**
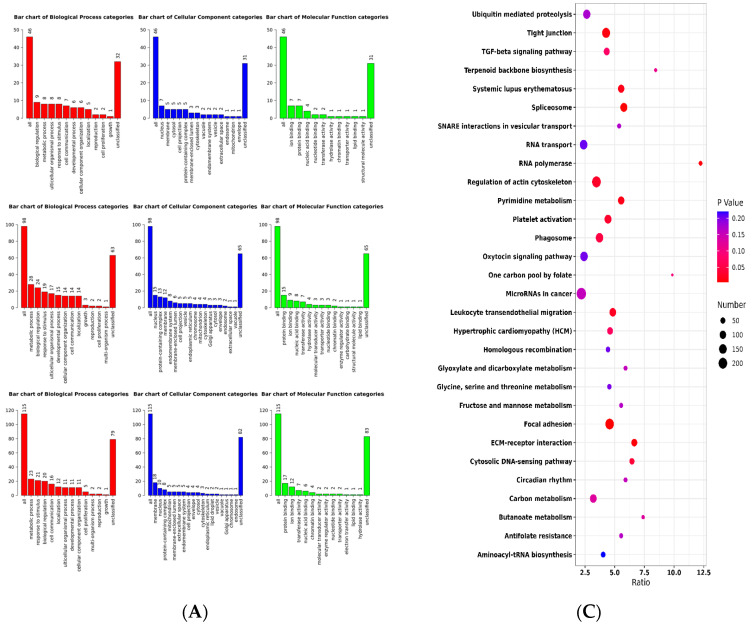
Kyoto Encyclopedia of Genes and Genomes enrichment analysis. (**A**) GO analysis of three Large White pig populations; (**B**) KEGG pathway analysis of the Canadian line; (**C**) KEGG pathway analysis of the American line; (**D**) KEGG pathway analysis of the Danish line. In Graph (**A**), the abscissa represents the GO terms that were the most enriched; the ordinate represents the number of genes that were enriched in this classification. The size of the circles represents the number of genes contained in the particular class in the graph (**B**–**D**), the larger the circle is, the more genes there are. Differently coloured circles represent the enrichment degree of false positives, the redder the circle is, the lower the false positive rate.

**Table 1 ijms-24-12914-t001:** Candidate genes are located in genomic regions based on selection signatures detection.

SSC (Sus Scrofa Chromosome)	Position (Mb)	Distance (bp) *	Genes
1	182.24–182.36	Upstream 117,606	*STYX*
2	2.79–2.91	Upstream 74,339	*SHANK2*
4	83.49–83.61	Upstream 4395	*CD247*
5	71.79–71.91	Upstream 10,224	*LRRK2*
9	130.54–130.66	Upstream 1660	*FLVCR1*
12	16.79–16.91	Downstream 14,536	*MYL4*
	54.85–55.35	Upstream 49,706	*MYH3*
13	129.24–129.36	Upstream 83	*FGF12*
14	131.14–131.216	Upstream 41,712	*FGFR2*
17	10.44–10.56	Downstream 3575	*SFRP1*
	10.64–10.76	Upstream 118,817	*miR-486*

* The distance was calculated as follows: The starting coordinate of the gene minus the starting coordinate of the selective signature region; candidate genes are a part of sequences located in the region.

**Table 2 ijms-24-12914-t002:** List of the top 1% runs of homozygosity was detected in three Large White pig populations and the overlapping QTL in pigQTLdb (https://www.animalgenome.org/cgi-bin/QTLdb/SS/index/; accessed on 16 March 2023).

Groups	Chromosome	Start (bp)	End (bp)	Length (bp)	Number of SNPs	pigQTLdb
American	1	43,045,542	43,357,270	311,728	5	-
		146,085,059	148,974,102	2,889,043	42	S: 145,869,313 E: 173,242,773 Average daily gain
	4	98,912,988	102,212,696	3,299,708	51	-
	6	102,107,540	102,337,903	230,363	4	-
		102,717,110	102,796,136	79,026	2	-
		102,917,556	104,370,915	1,453,359	23	-
		104,981,850	110,109,305	5,127,455	121	-
	7	72,338,896	74,402,073	2,063,177	42	S: 72,215,870 E: 87,765,126 Fat area percentage in carcass
	14	98,912,988	102,212,696	3,299,708	43	-
Canadian	4	44,727,463	48,379,816	3,652,353	54	S: 44,723,094 E: 91,039,884 Ham weight
	6	105,047,268	107,701,419	2,654,151	64	-
	7	50,032,121	52,080,918	2,048,797	31	-
		70,691,786	74,260,534	3,568,748	66	S: 70,292,251 E: 83,677,435 Teat number
	8	22,032,465	23,585,036	1,552,571	29	-
		24,568,718	25,096,226	527,508	2	S: 24,414,300 E: 25,683,843 Umbilical hernia
		57,175,682	58,758,032	1,582,350	35	S: 56,966,700 E: 67,491,976 Hematocrit
	14	92,778,821	94,149,712	1,370,891	39	-
Danish	2	71,619,091	74,328,649	2,709,558	33	S: 71,416,758 E: 128,795,277 Leaf fat weight
	5	51,535,641	54,004,977	2,469,336	52	-
		54,091,881	54,253,706	161,825	5	S: 54,354,525 E: 54,411,945 uterine horn length
	6	105,105,811	107,369,304	2,263,493	57	-
	9	83,483,921	88,168,152	4,684,231	111	S: 80,796,751 E: 97,479,874 Immunoglobulin G level
	13	86,541,183	88,583,284	2,042,101	48	S: 86,471,446 E: 118,227,339 Lean meat percentage
	15	76,452,937	76,592,155	139,218	5	S: 76,167,178 E: 76,761,699 Intramuscular fat content
		77,669,772	79,049,130	1,379,358	26	S: 77,173,290 E: 90,664,324 Drip loss

**Table 3 ijms-24-12914-t003:** Candidate genes related to the economic traits located in genomic regions with a high frequency of ROH.

SSC (Sus Scrofa Chromosome)	Start (bp)	End (bp)	Distance(bp) *	Genes
American line				
4	98,912,988	102,212,696	Upstream 11,008	*PLEKHO1*
6	102,917,556	104,370,915	Upstream 770,187	*LPIN2*
Meta-analysis				
6	105,105,811	107,369,304	Upstream 308,820	*ADCYAP1*
			Upstream 634,739	*CETN1*
			Upstream 934,231	*THOC1*
			Upstream 982,113	*USP14*
			Upstream 1,314,132	*GREB1L*
			Upstream 1,893,085	*miR-133*
			Upstream 2,177,038	*GATA6*

* The distance was calculated as follows: The starting coordinate of the gene minus the starting coordinate of the selective signature region; candidate genes are a part of sequences located in the region.

## Data Availability

No applicable.

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
