# Peer review of "Genomic Scan for Runs of Homozygosity and Selective Signature Analysis to Identify Candidate Genes in Large White Pigs"

_ijms, 2023, doi:10.3390/ijms241612914_

Round 1

Reviewer 1 Report

The work done by the authors looking for selection signatures on 2295 Large white pigs of Canadian, Danish and American lines is well organized and presents numerous analyses also well conducted. 

In my opinion, however, in its present form it presents insufficiently connected results.

On the one hand, the authors used Fst and LSBL as methods to identify selection signatures. Then they also carried out ROH analyses again looking for regions with high ROH frequencies under common selection in the three groups analyzed.

The main reccomandations are:

1) explain the authors what selection guidelines the Danish American and Canadian lines have adopted over time, instead of using the general phrase in the introduction line 35 "...aligning their characteristics with human requirements" and whether these selection lines have been different or the same.

In addition, let the authors report the provenance of their samples, i.e., where the animals (they genotyped) were bred.

2) The authors compare groups of very different sizes from 1500 American lines to 295 Danish lines. This difference could have an effect on the analyses performed.  The authors are therefore asked to comment on how this numerical difference might have affected their results and to verify by comparing a similar number of subjects representing the variability of the groups whether the identified regions and genes are confirmed.

3) They also explain the greater closeness of the groups reported in red and black on the pca and comment on whether subjects in the red group very close to the black group were used or eliminated in the analyses.

2) The authors remind that multiple methods must be used for selection signatures analyses. However, they use Fst together with LSBL the latter is a useful method for highlighting population substructures and is based on FSt. So this two methods are not independent. So the question arises whether it would not be more correct  and useful  to put together the results of FSt (top 1% of the signals) and top ROHs to highlight the regions under sleection in the different large white groups. Instead to present the results in an independetn way as they did.

It might then be useful to review the pooled results. 

Moreover, as the author commented that there are no evidence looking at the ROH distribution of peculiar differences in the three group of animals, in my opinion, it would be interesting to evaluate this statement also using FSt and in this case using LSBL

LSBL should be used also by investigating the selective neutral expectation of the investigated regions.

Reviewer 2 Report

I have several comments:

1) I could not understand why did the authors used three different populations. There seems no explanation/discussion in the manuscript. As also the authors denoted, the history of the population could affect the results of ROH analysis; however, the auhtors did not explain about the populations. The authors should add the description about this point throughaut the manuscript. For instance, how about the pedigree structure (e.g., close relatives were removed?)? How did each population was genetically improved? How about the impact of the number of samples used (especially low number for Danish)? How degree of ROH regions (or, broadly says, haplotypes) were shared among the populations? etc.

2) The authors should also discuss about false negatives for ROH detection. SNP marker density was enough, or more markers (such as NGS results) would be better? How about the impact of genotyping error and the setting of quatily control and ROH analysis? etc.

3) Was it suitable to calculate the distance in Table 1? I could not understand the negative distance.

4) I suspect whether the candidated genes reported in this study were truely "novel" findings. The authors must compare the previously reported candidate genes/regions for economically-important traits in pigs with the results in this manuscript, e.g., by using pigQTLdb.

Spell check would be encouraged throughout the manuscript.

Round 2

Reviewer 1 Report

The authors add  comments as suggested in the previous review,even if they could better integrate the results obtained with the Fst and ROH methods used.

Anyway the paper could be accepted in the present form.

Author Response

The authors add  comments as suggested in the previous review,even if they could better integrate the results obtained with the Fst and ROH methods used.

Anyway the paper could be accepted in the present form.

Response:

Thanks for your precious suggestion. As we mentioned before, the two methods, ROH and FST, are based on different principles. To gain a more comprehensive understanding of the signals left on the genomes of the three populations, we then used these different directions of selection methods. Meanwhile, our comparison also reveals that these two methods have no overlapping or similar loci. So, it’s hard to integrate the results obtained with the Fst and ROH methods.

Reviewer 2 Report

Major comments:

I could not understand the differences in the results among the three populations would be due to the difference in breeding scheme or the difference in the content of base populations (or both). The detailed discussion about this point should be strongly required.

Specific comments:

>2. How did each population was genetically improved?

>Response:

>Thanks for your precious suggestion. Added as suggested (Line 36- Line 42).

I think the added sentence is better for discussion, not introduction.

>4. How degree of ROH regions (or, broadly says, haplotypes) were shared among the populations? etc.

>Response:

Thanks for your precious question. On chromosome 6, from position 105105811 to 107369304, there is an overlap in the ROH of these three populations. 12% of the total ROH length in the American lines; 11% of the total ROH length in the Canadian lines; And 14% of the total ROH length in the Danish lines (Line 158- Line 161).

I could not understand why the authors focused on only chromosome 6. Please explain the reason in the manuscript.

Minor spell check might be required.

Author Response

  1. I could not understand the differences in the results among the three populations would be due to the difference in breeding scheme or the difference in the content of base populations (or both). The detailed discussion about this point should be strongly required.

Response:

Thanks for your precious suggestion. Added as the suggestion (Line 156- Line 165; Line 277; Line 279- Line 280; Line 289- Line 291).

Specific comments:

  1. How did each population was genetically improved?

>Response:

>Thanks for your precious suggestion. Added as suggested (Line 36- Line 42).

I think the added sentence is better for discussion, not introduction.

Response:

Thanks for your precious suggestion. Revised as suggested (Line 263- Line 270).

  1. How degree of ROH regions (or, broadly says, haplotypes) were shared among the populations? etc.

>Response:

Thanks for your precious question. On chromosome 6, from position 105105811 to 107369304, there is an overlap in the ROH of these three populations. 12% of the total ROH length in the American lines; 11% of the total ROH length in the Canadian lines; And 14% of the total ROH length in the Danish lines (Line 158- Line 161).

I could not understand why the authors focused on only chromosome 6. Please explain the reason in the manuscript.

Response:

Thanks for your precious question. We've made comparisons and found the overlap on chromosome 6 only (Line 152 – Line 155).

Round 3

Reviewer 2 Report

the manuscript has been improved.